# Milk-Derived Proteins and Peptides in Head and Neck Carcinoma Treatment

**DOI:** 10.3390/biom12020290

**Published:** 2022-02-10

**Authors:** Theresa Wang, Xinyi Liu, Yah Ying Ng, Kiera Tarleton, Amy Tran, Thomas Tran, Wen Yue Xue, Paul Youssef, Peiyu Yuan, Daniel Zhang, Rita Paolini, Antonio Celentano

**Affiliations:** Melbourne Dental School, The University of Melbourne, 720 Swanston Street, Carlton, VIC 3053, Australia; t.wang61@student.unimelb.edu.au (T.W.); x.liu119@student.unimelb.edu.au (X.L.); y.ng22@student.unimelb.edu.au (Y.Y.N.); k.tarleton@student.unimelb.edu.au (K.T.); amyuyen.tran@student.unimelb.edu.au (A.T.); thomas.tran@student.unimelb.edu.au (T.T.); w.xue6@student.unimelb.edu.au (W.Y.X.); paul.youssef@student.unimelb.edu.au (P.Y.); peiyu.yuan@student.unimelb.edu.au (P.Y.); zhangd4@student.unimelb.edu.au (D.Z.)

**Keywords:** head and neck, cancer, treatment, milk-derived peptides, proteins, human α-lactalbumin made lethal to tumour cells, HAMLET

## Abstract

Research investigating milk-derived proteins has brought to light the potential for their use as novel anticancer agents. This paper aims to systematically review studies examining the effectiveness of milk-derived proteins in the treatment of head and neck cancer. A systematic literature search of Medline, Evidence-Based Medicine, and Web of Science databases including papers published from all dates was completed. Inter-rater reliability was high during the title, abstract, and full-text screening phases. Inclusion criteria, exclusion criteria, and data extraction were based on the PICOS tool and research questions. Reporting followed the Preferred Reporting Items for Systematic Review and Meta-Analysis criteria. Eligible in vitro and in vivo studies (*n* = 8/658) evaluated lactoferrin, α-lactalbumin, and its complexes, such as HAMLET, BAMLET and lactalbumin-oleic acid complexes, as well as lactoperoxidase, whey, and casein. Their effectiveness in the treatment of head and neck cancer cells lines found that these compounds can inhibit tumour growth modulate cancer gene expression, and have cytotoxic effects on cancer cells. However, the exact mechanisms by which these effects are achieved are not well understood. Systematically designed, large, optimally controlled, collaborative studies, both in vitro and in vivo, will be required to gain a better understanding of their potential role in the treatment of head and neck cancer.

## 1. Introduction

Head and neck cancer is an umbrella term that collectively describes cancers in the oral cavity, nasal cavity, salivary glands, sinuses, pharynx, or larynx. Globally, they account for more than 650,000 cancer cases and 330,000 deaths each year [1]. The most common head and neck cancer cell types are squamous cell carcinomas, which originate from the outermost layers of the mucosa. Specifically, oral squamous cell carcinomas (OSCC) account for up to 90% of oral malignant tumours [2]. Despite new treatment modalities, there is a persistent 19–59% 10-year survival rate for head and neck cancers [3]. The main treatments currently available for OSCC include surgery, radiation therapy, medications, or a combination of the three. All possess numerous side effects and complications, as well as questionable levels of efficacy [4]. As such, cancer research is constantly looking for safer and more effective treatment options.

The research investigating milk-derived proteins has highlighted the potential for their use as novel anticancer agents. Milk proteins can be classified into two key groups: caseins and whey proteins [5]. Caseins are grouped as α-casein, β-casein, and κ-casein; whey proteins refer to a variety of proteins including alpha-lactalbumin, lactoferrin, beta-lactoglobulin (in bovine milk), and other biologically active proteins, such as enzymes, mineral-binding proteins, and immunoglobulins [5]. Numerous studies have indicated that these milk-derived proteins and peptides may be a safe and effective adjuvant to conventional medical therapies. Specifically, human alpha-lactalbumin made lethal to tumour cells (HAMLET), has been shown to exhibit therapeutic anticancer effects in several animal models including bladder cancer, glioblastomas, and intestinal cancer and in human studies against skin papilloma and bladder cancer [6,7,8,9,10]. Many of these components have been used as ingredients of nutraceuticals and hygiene products; however, research investigating the pharmaceutical applications of milk-derived proteins and peptides is still very limited.

There is growing interest in milk-derived proteins and their anticancer effects on head and neck squamous cell carcinomas (HNSCC). This systematic review aims to present the current research surrounding milk-derived proteins and their effects. Specific questions addressed in this review included:1.How effective is the use of milk-derived proteins as an anticancer treatment for HNSCC?2.What are the anticancer mechanisms of action behind these milk-derived proteins?3.How are the milk-derived proteins extracted, obtained, and synthesised?4.What is the specificity of milk-derived proteins in targeting HNSCC vs. normal host cells?5.Are the effects of milk-derived proteins cell line, dose, and time dependent?

## 2. Materials and Methods

The systematic review was collated according to the Preferred Reporting Items for Systematic Reviews and Meta-Analyses (PRISMA) statement [11].

### 2.1. Study Selection

#### 2.1.1. Inclusion Criteria

The inclusion criteria were formulated based on the following PICOS tool questions:
P = Patients/population/case: HNSCC.I = Intervention: Milk-derived proteins as the sole treatment or cotreatment.C = Control: Specified control in each study.O = Outcome: Prognosis of head and neck carcinoma.S = Study design: All original studies including in vitro and/or in vivo and/or in silico.


Eligible studies included those that were published at any point in time.

#### 2.1.2. Exclusion Criteria

Studies were excluded if they were:1.Published only in non-English languages or if the full text was not available.2.Irrelevant to the association of milk-derived proteins and HNSCC.3.Involving cancers that occur in head and neck regions but are not usually classified as head and neck cancers, including cancer of the scalp, skin, muscles, bones, eye, brain, ear, oesophageal, thyroid, and parathyroid.4.Focusing on prevention of cancer rather than on the treatment of cancer.5.Exploring the role of milk-derived proteins as a drug delivery vehicle.

#### 2.1.3. Screening Process

A literature search and screening process were then undertaken to identify those studies eligible for inclusion in this systematic review, and this process is summarised in the PRISMA flowchart (Figure 1).

Step 1.Electronic literature searches using the workgroup-defined search strategies were conducted by two blind reviewers (DZ, WX) on 9 July 2021 in Medline (Ovid), Evidence-based Medicine (EBM), and Web of Science databases. The syntaxes for these searches are shown in Appendix B. The results were imported into Endnote X9 (Clarivate Analytics, Philadelphia, PA, USA); automated deduplication was completed by Endnote. The subsequent library was exported to Covidence (Veritas Health Innovation, Melbourne, Australia) for screening, where another automated deduplication was performed.Step 2.Papers were then screened by title only.Step 3.Papers were then screened by abstract.Step 4.Papers were lastly screened by full text.

### 2.2. Statistical Analysis

Data were extracted from the final articles eligible for inclusion in this systematic review and tabulated. Cohen’s kappa statistic for inter-rater agreement and absolute percentage agreement were calculated using IBM Statistics 27 (SPSS).

Heterogeneity of the studies and the variability in methods used to assess outcome precluded the performance of any further quantitative analyses, such as meta-analyses.

### 2.3. Risk of Bias

The 8 studies included in this systematic review were assessed via the Office of Health Assessment and Translation (OHAT) risk of bias tool for internal validity. The OHAT tool is comprised of 6 main domains and 11 questions that allow for the assessment of human and animal experimental studies. In addition to the 6 main domains, there are ‘other biases’ that may be evaluated depending on relevance. For the studies in this systematic review, it was deemed that 5 domains ‘selection’, ‘performance’, ‘attrition/exclusion’, ‘detection’ and ‘selective reporting’; other biases ‘statistics’ and ‘unintended co-exposures for experimental studies’ were relevant for the assessment. As such, 10 questions of the OHAT tool were assessed to be relevant in the evaluation of the 8 included studies. The domains and questions relevant to each domain included are shown below. Each question under each domain included was rated with risk of biases of “definitely low”, “probably low”, “probably high”, and “definitely high”.

Selection bias:Was the administered dose or exposure level adequately randomized?Was the allocation to study groups adequately concealed?

Performance bias:Were the experimental conditions identical across study groups?Were the research personnel and human subjects blinded to the study group in the study?

Attrition/Exclusion bias:Was the outcome data complete without attrition or exclusion from the analysis?

Detection bias:Can we be confident in the exposure characterization?Can we be confident in the outcome assessment?

Selective reporting bias:Were all the measured outcomes reported?

Other biases:Statistics: Were the statistical methods appropriate?Unintended coexposures for experimental studies: Did the study design or analysis account for important confounding and modifying variables (including unintended coexposures) in experimental studies?

## 3. Results

### 3.1. Study Selection

The process to select eligible studies is illustrated in (Figure 1). Out of the 658 papers initially found during the literature search, 157 papers were removed during the automated deduplication process (performed by Endnote and Covidence), leaving 501 unique papers to undergo screening.

Cohen’s kappa statistic [κ] for inter-rater agreement and absolute percentage agreement for each step of the screening process, respectively, was 0.84 (95% confidence interval [CI]: 0.75–0.93) and 97.6% for step 2; 0.89 (95% CI: 0.74–1.04) and 95.0% for step 3; and 1.0 (95% CI: 1.0–1.0) and 100% for step 4. The reasons for the exclusion of these papers are outlined in the inclusion and exclusion criteria. After the screening process, eight papers were identified to be included in this systematic review.

### Study Characteristics

The eight papers were selected from a timespan from August 2003 to June 2020 and originated from multiple countries. All eight articles examined milk-derived proteins and peptides targeting HNSCC cells in vitro, with just one study also including an in vivo study (Table 1).

### 3.2. Risk of Bias

Risk of bias assessment via the OHAT tool evaluated domains of bias relevant to included studies. The risk of bias was found to be “definitely low” in domains ‘selection’, ‘performance’, ‘attrition/exclusion’, ‘detection’, and ‘other biases’ (87.5%, 75%, 62.5%, 75%, and 68.75%, respectively) (refer to Figure 2). A bias rating of “probably low” was found in domains ‘performance’, ‘detection’, and ‘selective reporting’ (12.5%, 12.5%, and 62.5%, respectively). The risk of bias was found to be “probably high” in domains ‘selection’, ‘performance’, ‘attrition/exclusion’, ‘detection’, and ‘other biases’ (12.5%, 6.75%, 12.5%, 6.75%, and 25%, respectively). A “definitely high” risk of bias rating was found for domains ‘performance’, ‘attrition/exclusion’, ‘detection’, ‘selective reporting’, and ‘other biases’ (6.75%, 25%, 6.75%, 37.5%, and 6.25%, respectively). Refer to Appendix A for the full OHAT risk of bias assessment.

### 3.3. Specimen Types

All eight papers were in vitro studies that used cell lines, and one included an in vivo portion of the study, in which immunocompetent mice were used. In total, 16 cell lines were used over the eight studies. The cell line O12 from Wolf et al. (2003) [12] was later found to be genetically identical to O22 [13]. Therefore, after discounting the repeats there were nine different human cancer cell lines used (Table 2). This covers cells from the buccal mucosa (1/9), tongue (3/9), gingiva (1/9), larynx (2/9), hypopharynx (1/9), and oral cavity (1/9). Three mice models were used, and mice squamous cell carcinoma (SCCVII) and fibrosarcoma (RIF) cell lines were injected into the neck or floor of the mouth.

### 3.4. Source of Milk-Derived Proteins and Peptides

The proteins and peptides used for these studies were either obtained from external sources or made in the laboratory. Since these proteins and peptides can be derived from milk products, two papers used commercial products to extract the required molecules and one approached the milk industry. For HAMLET/BAMLET (3/8), they were prepared in the lab following the procedure as previously described [16,17,18]. Several papers also obtained their products from chemical companies/laboratories.

### 3.5. Milk-Derived Proteins and Peptides Examined

Amongst the eight papers included for this systematic review, seven milk-derived proteins and peptides were examined (Table 3). A detailed description of them is reported below.

#### 3.5.1. Whey and Casein

Panahipour et al. (2021) [20] showed that whey and casein powders processed from milk remain rich in TGF-β and supposedly lactoperoxidase (LP). The paper primarily focused on the effects of the powders on TGF-β and DNA-binding protein inhibitors (ID). It was found that whey and casein decreased the expression of ID1 and ID3 genes in oral squamous carcinoma cells. ID genes are overexpressed in OSCC and may serve as a prognostic factor; however, a clinical translation of decreasing the ID genes is unknown [20]. The powders also enhanced TGF-β target genes, which mediate the increase of IL11, NOX4, PRG4, smad phosphorylation, and nuclear translocation [20]. Whilst this may have some health benefits, there were no results of TGF-β influencing HSC2 target genes.

#### 3.5.2. Lactoferrin (LF)

Lactoferrin is an iron-binding glycoprotein that is found at high concentrations in breast milk. A total of three studies (two in vitro, one both in vitro and in vivo) examined the anticancer effects of lactoferrin. Amongst these three papers, two used bovine lactoferrin while one did not specify its source. All of the papers reached a consensus that lactoferrin alone has no cytotoxic effect on OSCC lines [12,21,22]. However, lactoferrin was shown to be able to reduce tumour growth in immunocompetent (CH3/HeJ) mice but not in athymic nude nu/nu mice meaning it might have an immunomodulatory effect [12]. Furthermore, when combined in a 1:2 ratio with tea polyphenol-E (P-E) it was found to significantly enhance the cytotoxic effect of P-E on CAL-27 and human fibroblasts cells in vitro [21].

#### 3.5.3. Pepsin-Digested Lactoferrin (pLF)

Sakai et al. (2005) [22] examined the anticancer potential of pLF. pLF was found to induce apoptotic cell death on human tongue-derived SCC cell lines (SAS) in a dose-dependent manner, possibly via the JNK/SAPK pathway.

#### 3.5.4. Lactoperoxidase (LP)

LP is an active antimicrobial agent found in milk and dairy products that is heat and pH stable [20]. Panahipour et al. (2020) [23] identified that LP was able to decrease the expression of ID1, ID3, and Distal-less Homeobox 2 (DLX2) genes in HSC2 cells but not in normal cells. The ID genes are involved in cell growth regulation and are therefore upregulated in various cancers and associated with tumour angiogenesis. It is theorised that LP may lower reactive oxygen species and thus decrease the expression of ID3 as it is a redox-sensitive gene [23]. The MAPK pathway may also mediate its effects on HSC2 cells.

#### 3.5.5. Alpha-Lactalbumin (α-LA)

A total of three studies, all in vitro, assessed the cytotoxic potential of α-LA. α-LA is found naturally in the whey portion of milk and was found to have minimal cytotoxic effects on cancer cells for concentration of up to 0.1 mM [24,25,26]. α-LA was also found to have differential toxicity on different cell lines. A higher reduction in cell viability was observed in the dysplastic cell line DOK 24 h after treatment [24].

#### 3.5.6. BAMLET/HAMLET

Amongst the eight papers included in this systematic review, one study focused on BAMLET [24] and two focused on HAMLET [25,26], both evaluating their cytotoxic effects on cancerous cell lines. All of the studies unanimously agreed that oleic acid (OA) was the main cytotoxic component of BAMLET/HAMLET, while alpha-LA serves as a carrier molecule to enhance cellular delivery [24,25,26]. Also, cytotoxic effects of BAMLET/HAMLET were found to be both cell type and dose dependent. BAMLET was determined to be cytotoxic against both oral dysplastic and cancer cells possibly via cell cycle arrest and apoptosis [24]. The articles included in this systematic review only examined the anticancer properties of HAMLET against larynx carcinoma cells, which has been proven to be effective [25,26].

#### 3.5.7. Alpha Lactalbumin-Oleic Acid (LA-OA) Complexes

A total of two studies, both in vitro, discussed the preparation and anticancer potential of the LA-OA complexes against squamous cell carcinoma. LA-OA complexes were found to have similar structural and cytotoxicity properties to HAMLET, with the LA-OA-45 state determined to be the most analogous to HAMLET. Also, the conventional protocol used of ion-exchange chromatography for the preparation of HAMLET was not a requirement for forming these complexes. LA-OA complexes can be formed by adding OA to protein solution instead [25,26].

## 4. Discussion

The efficacy of milk-derived proteins on HNSCC has been modestly investigated in the literature with several studies exploring their effects on genes and malignant cell lines. A common finding through the available studies is that milk-derived proteins alone do not induce cytotoxic effects in HNSCC cells, however, when complexed with an active component, they have the ability to synergistically and selectively target cancerous cells.

Several studies demonstrated that both BAMLET/HAMLET and LA-OA complexes possess similar cytotoxicity against HNSCC cells [24,25,26]. When not in a complex, α-LA did not demonstrate any cytotoxic effect against OSCC cells in vitro; however, it was shown to enhance the cellular delivery of OA as the BAMLET complex. This is supported by evidence that the IC_50_ value of OA alone is twice that of BAMLET [24]. Similarly, LF was found to possess no cytotoxicity against HNSCC cells in vitro; however, in combination with P-E in a 2:1 ratio (40 µg LF + 20 µg P-E), it significantly increased the cytotoxic effect of P-E against CAL-27 cells [21]. Sakai et al. (2005) also discovered that 0.125 to 2.0 mg/mL of lactoferrin digested by pepsin induced apoptotic cell death in SAS cells in vitro and thus may be effective in oral mucosal tumour immunity and prevention [22].

A desirable feature of any anticancer drug is the ability to selectively target cancer cells while sparing host cells. Numerous studies provided promising evidence on the ability of milk-derived proteins to selectively target cancer cells [18,27,28]. Mohan et al. (2007) demonstrated that tea polyphenols with added bovine LF milk protein preferentially exerted cytotoxic effects on human tongue squamous carcinoma cells (CAL-27) compared to noncancerous human gingival fibroblast cells [21]. Similarly, research by Panahipour et al. (2020) found that lactoperoxidase, a 5% aqueous fraction of human milk, cow’s milk, and infant formula decreased the expression of ID-1, ID-3 and DLX2 genes in HSC2 cells but not in the gingival fibroblasts [23]. Despite these findings, little is known about how this selective cytotoxicity is achieved.

At present, the exact anticancer mechanisms of these milk-derived proteins and peptides remain unclear; however, numerous theories have been proposed to explain the effect. Sinevici et al. (2021) suggested that the mechanism of BAMLET cytotoxicity may involve multiple simultaneous events including cell cycle arrest, autophagy, and a minor involvement of necrosis [24]. Autophagy and cell cycle arrest were observed in both oral cancer cells as well as dysplastic cells, with the latter found to be more sensitive to BAMLET [24]. The suggested initiation of cytotoxic mechanisms of BAMLET/HAMLET include membrane integration, ion fluxes, and inhibition of kinase proteins, as well as high affinity to histones, which disrupt chromatin assembly [25,29]. It has been shown to interact with the mitochondria, endoplasmic reticulum, proteasomes, and lysosome once inside the cell [29].

Sakai et al. (2021) demonstrated that Lfn-p was able to induce phosphorylation of extracellular signal-regulated kinase (ERK1/2) and c-Jun N-terminal kinase/stress-activated protein kinase (JNK/SAPK) [22]. Both belong to the MAP kinase signalling pathways and play a key role in cell growth, survival, differentiation, and apoptosis. Thus, the dose-dependent treatment with bovine pLF on SAS cells caused phosphorylation of ERK1/2 and JNK/SAPK proteins, caspase-3 activation, and several morphological changes [22]. This involved membrane blebbing, cellular shrinkage, apoptotic bodies, nuclear condensation, and fragmentation of the DNA [21,22]. Furthermore, lactoferrin was able to cause apoptosis by producing reactive oxygen species (ROS) in human leukaemia cell lines in vitro [22]. As a result, they hypothesised that this increase in ROS may result in damage to cellular structures, such as proteins, DNA, lipids, membranes, and organelles, ultimately leading to cell death. In addition, Mohan et al. (2007) stated that the cotreatment with P-E and LF induced apoptosis in the G1 phase of the cell cycle also via ROS controlled by Bcl-2 and Bax proteins. Bcl-2 is a major antiapoptotic protein and is upregulated in cancers, whilst Bax is proapoptotic [21]. P-E and LF decreased the Bcl-2/Bax ratio, which caused apoptosis through the generation of ROS, which increased the mitochondrial transmembrane potential and activated the caspase cascade [21].

The immune system may also be involved in milk-derived proteins combating HNSCC. Although LF was shown to have no effect against HNSCC cells in vitro, intratumoural injection of 250 µg LF for 4 days into C3H/HeJ mice with SCCVII or RIF cell lines was shown to significantly reduce tumour growth by approximately 50% when compared with controls [12]. However, since there was no significant reduction in the athymic mice, it was hypothesised that LF might inhibit tumour growth via immunomodulation, specifically by upregulation of T-cells [12]. Another theory is that LF may inhibit tumour cells from secreting IL1-α, which causes increased tumour growth [12].

The papers included in this review explored a few specific genes and genetic profiles; however, due to a lack of evidence as well as the heterogeneity of the cell lines and milk-derived proteins used, a correlation could not be established between genetic mutation and anticancer effect. However, the anticancer properties of BAMLET were determined to be p53 independent, indicating that BAMLET may potentially exhibit therapeutic potential in both OSCC associated with p53 mutations and those that do not [24]. Reconstructed casein and whey powder were also shown to maintain TGF-β activity in gingival fibroblasts, while decreasing the expression of ID-1 and ID-3 in human OSCC lines by approximately 10 time and 17.5 times, respectively [23]. ID genes are highly implicated in tumour growth and angiogenesis, while TGF-β is a tumour suppressor protein that is important in the initial stages of tumourigenesis [30,31]. Panahipour et al. (2021) also showed that the high levels of TGF-β in whey and casein protein powders led to the phosphorylation of SMAD3 and nuclear translocation of SMAD2/3 [20]. SMADs are known to bind to DNA and regulate proliferation, differentiation, and apoptosis [32]. Additionally, Panahipour et al. (2020) demonstrated that the expression of ID-1 and ID-3 genes in OSCCs were decreased by around five and four folds, respectively upon exposure to 100 µg/mL lactoperoxidase [23]. This suggests the capability of milk proteins to reduce the levels of transcription factors in OSCC cells.

Almost all papers showed that the effects of milk-derived proteins on HNSCC cell lines are dose and time dependent. In vitro studies of milk-derived proteins on OSCC cells demonstrated that the effect of cytotoxicity on OSCC cells increased as dose and duration of treatment increased. Sinevici et al. (2020) studied the time and dose dependency effects of BAMLET on DOK, Ca9.22, and TR146 cell lines. In all cell lines, cell death was visible within the first hour, with complete cell death achieved within 18 h [24]. However, specific data regarding this temporal relationship was not provided in the paper. Sakai et al. (2005) demonstrated that bpLF-treated SAS cell lines induced cell necrosis in a dose-dependent manner, where the release of LDH increased as the dose of bpLF increased (0.125 to 2.0 mg/mL) [22]. Furthermore, Wolf et al. (2003) demonstrated that tumour volume in C3H/HeJ mice with SCCVII and RIF cell lines treated with intratumoural injection of lactoferrin decreased as the number of doses increased [12]. Time and dose dependency of milk-derived proteins play an important role in determining the therapeutic doses and duration of anticancer treatment.

A variety of methods for the extraction and synthesis of milk-derived proteins were utilised by each of the studies included in this review. Casein and whey proteins are both derived from bovine milk. Casein can be formed by adding either hydrochloric acid or sulphuric acid to milk, causing the casein micelles to coagulate [33]. This process forms acid casein, which can be neutralised, dried, and milled to produce casein powder [20]. After the acidification, the remaining liquid contains the soluble whey protein. In the study conducted by Panahipour et al. (2021), casein and whey protein powders were commercially obtained [20].

Most of the studies that investigated lactoferrin obtained it from chemical companies including Wako Pure Chemical Industries, Ltd., Osaka and Morinaga Milk Industry Co. Ltd., Tokyo Japan [21,22]. Only one study examined lactoperoxidase, and they purified this peptide from bovine milk via size fractionation due to its molecular weight and biological activity [23].

Permyakov et al. (2011) and Knyazeva et al. (2008) isolated their α-LA via gel filtration and anion-exchange chromatography as described in Neyestani et al. (2003) [34] and via procedures detailed in Kaplanas et al. (1975) [16].

HAMLET and BAMLET are relatively novel milk-derived complexes. The conventional protocol for preparing these two complexes is ion-exchange chromatography [24,26]. HAMLET is produced in the same way but with human milk-derived α-LA. In contrast to HAMLET/BAMLET, the preparation of HAMLET-like hLA-OA complexes does not require the complicated conventional protocol. These complexes were prepared by titration of a diluted solution of Ca2+- free hLA (ca. 10 μM; pH 8.3, 20 mM H3BO3-KOH, 150 mM NaCl, 1 mM EDTA; 25 mL) and 3-8 μL oleic acid with 96% ethanol (100mM) [25]. Approximately 10 titrations were required to achieve the effective critical micelle concentration value at 17 °C (for LA-OA-17) or 45 °C (LA-OA-45 state) [25,26]. The fact that LA-OA complexes can be prepared in an aqueous solution allows for a more transparent and controllable manufacturing process [25,26].

In conducting this systematic review, several limitations were identified. Firstly, the inclusion of only eight papers provided a limited pool of evidence from which conclusions could be drawn. Furthermore, due to the limited number, complexity, and heterogeneity of the included studies, the potential for quantitative synthesis of the findings, such as meta-analysis, was precluded. In addition to this, data investigating the effects of the treatment with milk-derived proteins on normal cells was limited, with only three papers utilising normal controls [20,21,23]. This raises the question of how specific these milk-derived proteins and complexes truly are selectively targeting HNSCC cells whilst sparing non-cancerous host cells.

Although the papers reported the occurrence of specific cytotoxic effects, the exact mechanisms by which milk-derived proteins exert these effects remain largely unknown. As such, future research is needed to investigate the specific pathways leading to the cytotoxic effects of milk-derived proteins and complexes, as well as the effect of using milk-derived proteins in combination with established chemotherapeutic agents. Additionally, there is a need for further in vivo and clinical studies, as well as studies that include control specimens as this will allow insight into the specificity and safety of milk-derived proteins and complexes. Furthermore, this field would benefit from investigation to maximise the efficiency of milk-derived protein complexes synthesis and development.

## Figures and Tables

**Figure 1 biomolecules-12-00290-f001:**
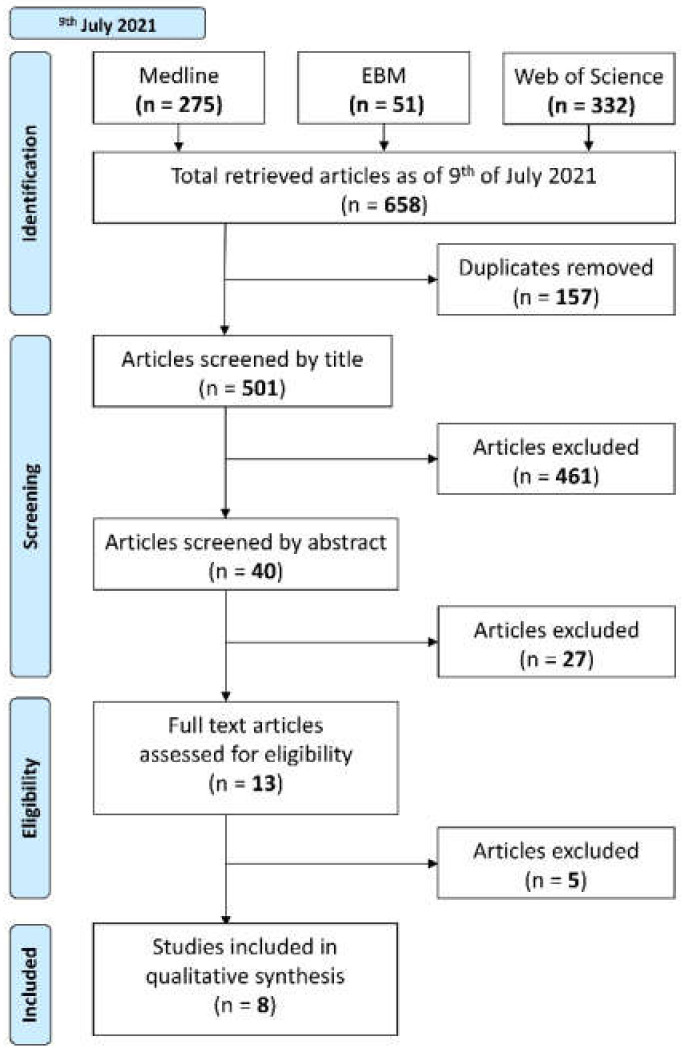
Selection of studies for the systematic review of milk-derived proteins and peptides in head and neck carcinoma treatment.

**Figure 2 biomolecules-12-00290-f002:**
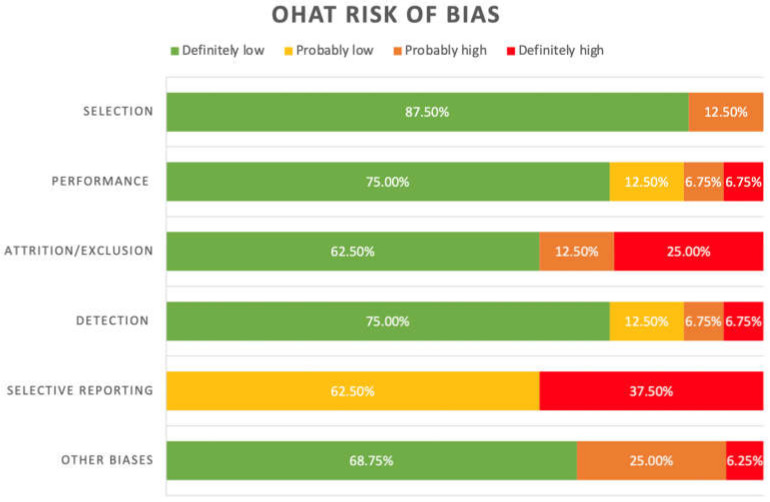
Summary of risk of bias assessment of included studies under the OHAT guidelines.

**Table 1 biomolecules-12-00290-t001:** Findings from 8 studies on milk-derived proteins and peptides on head and neck cancer cells.

Author, Year, Country	Major Proteins and Peptides	Model	Experimental Method	Source of Proteins and Peptides	Major Findings
Sinevici et al., 2020, Ireland	- BAMLET (0.0–1.0 mg/mL). - α-LA (0.0–0.1 M).	- DOK: Human oral keratinocyte. - TR146: Human OSCC. - Ca9.22: Human gingival SCC.	- In vitro. - AlamarBlue redox indicator. - LDH assay, flow cytometry, and confocal microscopy.	BAMLET: Ca^2+^ depleted α-LA added into DEAE column matrix preconditioned with OA then eluted with NaCl buffer, desalted overnight and lyophilized.	- BAMLET is cyto. to both dysplastic and OSCC cells; cell-type and time dependent. - Sensitivity of cells to BAMLET: DOK > Ca9.22 > TR146 (p53-independent; potential intervention for both OSCC subtypes). - OA: active component; IC50 is 2x > BAMLET. - α-LA: carrier molecule; transports OA and dissociates at cell membrane.
Wolf et al., 2003, USA	- LF (25, 50, 100, or 250 μg/day).	- O22: Human laryngeal SCC. - O12: Human laryngeal SCC. - FaDu: Human hypopharyngeal SCC.	- In vitro. - Cell count.	NR	- SCCVII or RIF: LF ↓ tumour growth. - O22: LF did not ↓ in tumour growth. - No difference in cell counts between control and SCC cell lines at all doses of LF. - LF inhibits malignant tumour growth via immunomodulation.
	- LF (250 μg for either 2,3, or 4 days): C3H/HeJ murine with RIF/SCCVII. - LF (100–500 μg for 4 days): Athymic nu/nu mice with O22.	- C3H/HeJ with SCCVII tumour: murine SCC. - C3H/HeJ with RIF tumour: murine fibrosarcoma. - Athymic nude nu/nu mice model with O22 tumour.	- In vivo. - Tumour volume.	NR	
Panahipour et al., 2020, Austria	- 5% AqF: pasteurized human milk, cow’s milk, reconstituted infant formula. - bLP (100 μg/mL). - bLF (100 μg/mL). - Recombinant human osteopontin (50 ng/mL).	- HSC2: Human OSCC. - TR146: Human OSCC.	- In vitro. - Microarray analysis. - MTT assay; real-time quantitative reverse transcription PCR analysis + immunoassay. - Western blot analysis.	- Human milk: Medical University of Vienna. - Dairy products: commercial. - All batches processed in centrifuge and their AqF heated before freezing.	- AqF human milk, cow’s milk, and infant formula ↓ ID1, ID3, and DLX2 expression in HSC2 cells; caused ID3 suppression in TR146 cells; not in HGF. - bLP: ↓ ID1, ID3, and DLX2 basal expression in HSC2 cells. - bLF and osteopontin: ↑ ID1 expression. - MAPK mediates milk-induced ID1, ID3, and DLX2 suppression in HSC2 cells.
Mohan et al., 2007, India	- bLF (40 μg/mL) as cotreatment with 20 μg/mL P-E and 40 μg/mL P-B.	- CAL-27: Human tongue SCC.	- In vitro. - MTT assay. - Fluorescent microscopy. - Oxidation-sensitive fluorescent probe. - Flow cytometric analysis + Cell Quest software. - Western blot analysis. - CASP-3-C colorimetric kit.	bLF: Morinaga Milk Industry Co. Ltd., Tokyo Japan.	- bLF: no cyto. effects. - P-E or P-B: dose-dependent cyto. effects. - Cotreatment of P-B and bLF: ↓ P-B cytotoxicity; bLF suppresses anticancer effects of P-B. - Cotreatment of P-E and bLF: synergistic cyto. effect against CAL-27, less on HGF.
Sakai et al., 2005, Japan	- bpLF (5% bLF + 3% pepsin).	- SAS: Human tongue-derived SCC.	- In vitro. - Cyto. activity based on lactate dehydrogenase (LDH) activity. - Fluorescence microscope; apoptosis Ladder Detection Kit + agarose gel electrophoresis. - ECL Western blot detection system. - Electrophoresis.	- bLF: Wako Pure Chemical Industries, Ltd., Osaka, Japan. - Pepsin: Difco Laboratories, Livonia, MI, USA. - bpLF: pepsin added into distilled water with bLF and incubated, heated and pH-adjusted. Precipitate was centrifuged and supernatant was retained and concentrated.	- bpLF: apoptotic cell death in SAS cells; induced phosphorylation of ERK1/2 and JNK/SAPK. - SAS cells treated with bpLF and JNK/SAPK inhibitor: ↓ cyto.ity. - SAS cells treated with bpLF and MEK1 inhibitors: ↑ cyto.ity.
Permyakov et al., 2011, Russia	- bLA-OA-45 (1–300 µM). - HAMLET (1–300 µM). - Intact bLA (1–300 µM). - Reference bLA (1–300 µM).	- HEp-2: Human epidermoid larynx carcinoma.	- In vitro. - Crystal violet assay.	- hLA: made according to Kaplanas and Antanavichyus (1975). - Ca^2+^ depleted bLA: Sigma-Aldrich Co, Moscow, Russia. - HAMLET and BAMLET: prepared using chromatography Ca^2+^-free α-LA preconditioned with OA and eluted with salt. - bLA-OA-45: synthesised using Ca^2+^-free bLA titrated with OA solution at 45 ℃. - Reference bLA: same procedures for bLA-OA-45 but skipping OA.	- HAMLET and bLA-OA-45: similar cyto.ity. - OA: active component; bLA: carrier of OA. - The hydrophobic effect in stabilisation of the α-LA-OA complex → possibility in preparing similar complexes possessing cyto.ity w.r.t. malignant cells.
Panahipour et al., 2021, Switzerland	- 0.1, 1.0, 10.0% AqF whey and casein.	- HSC2: Human OSSC.	- In vitro. - MTT assay, Ki67 and cyclinD1 markers. - Real-time quantitative reverse transcription PCR analysis of TGF-β1 gene. - IL11 Quantikine ELISA testing. - Western blot analysis. - Alexa Fluor^®^ 488-conjugated secondary ab; fluorescent microscope.	- Casein and whey powder: commercial. - AqF whey and casein: reconstituted with serum-free DMEM.	- Processing milk into casein or whey maintains its TGF-β and theoretically its LP activity - 1% AqF casein and whey powder did not affect viability and proliferation of HGF. - Casein and whey powder: ↑ TGF-β target genes which ↑ IL11, NOX4, PRG4 and phosphorylation of smad3. - Casein and whey powder: ↓ ID1 and ID3 expression in OSCC
Knyazeva et al., 2008, Russia	- HAMLET (0–1 mM). - hLA with OA (0–1 mM). - α-LA (0–1 mM).	- HEp-2: Human epidermoid larynx carcinoma.	- In vitro. - Crystal violets assay.	- hLA: prepared according to Kaplanas and Antanavichyus (1975). - OA: Aldrich. - HAMLET: prepared according to Pettersson et al. (2006). - LA-OA-17 and LA-OA-45: Ca^2+^ free hLA titrated with OA solution. Complexes then calcium dialyzed, lyophilized, and stored. - Ca^2+^ free hLA: prepared according to Blum et al.	- HAMLET: dose-dependent cyto.ity. - OA: toxic effect when >300 µm. - Intact hLA: no toxic effect up to 1 mM. - LA-OA-45 state: the most analogous to HAMLET.

>, greater, higher more than; ≥, greater or equal to; <, less, slower, lower than; ↑, increase; →, leads to, results in; ↓, reduce, decrease; /, per; α-LA, α-lactalbumin; ab, antibody; AqF, aqueous fraction; BAMLET, bovine α-lactalbumin made lethal to tumour cells; bLA, bovine α-lactalbumin; bLF, bovine milk lactoferrin; bLP, bovine milk-derived lactoperoxidase; bpLF, bovine pepsin-ingested lactoferrin; cyto., cytotoxic; cyto.ity, cytotoxicity; DEAE, diethylaminoethyl cellulose; DOK, dysplastic oral keratinocyte cell line; h, hour, hours; HAMLET, human α-lactalbumin made lethal to tumour cells; hLA, human α-lactalbumin; hLA-OA, human α-lactalbumin and oleic acid; hSCC, human squamous cell carcinoma; IC_50_, half-maximal inhibitory concentration; IL11, interleukin 11; LA-OA, α-lactalbumin and oleic acid complex; LDH, lactate dehydrogenase; LF, lactoferrin; LP, lactoperoxidase; mins, minutes; NOX4, NADPH oxidase 4; NR, not reported; nu/nu, nude; OA, oleic acid; OC, oral cancer; OEC, oral epithelial cell; OSCC, oral squamous cell carcinoma; P-B, Polyphenon-B; PCR, polymerase chain reaction; P-E, Polyphenon-E; PRG4, proteoglycan 4; ROS, reactive oxygen species; SCC, squamous cell carcinoma; w.r.t., with regards to; X, time.

**Table 2 biomolecules-12-00290-t002:** Properties of HNSCC cell lines: tissue of origin, genetic characteristics and mutations, and notable characteristics from the 8 studies.

Cell Lines	Tissue of Origin	Genetic Characteristics and Mutations [14]	Notable Characteristics [14,15]
DOK	Tongue	TP53 mutation	Mild to moderate dysplastic Stratification in confluent cultures and contain a keratin profile similar to the original dysplasia Nontumourigenic in nu/nu mice
TR146	Buccal mucosa	p53 wild type	Well-differentiated Polygonal Tumourigenic in female (nu/nu) mice Metastatic
Ca9.22	Gingiva	p53 mutation	Tumourigenic Epithelial-like Expressing remarkable EGF receptors
O22	Larynx	TP53 mutation	Metastatic
O12	Larynx	TP53 mutation	Metastatic
FaDu	Hypopharynx	CDKN2A, FAT1, or TP53 mutation	Tumourigenic in nu/nu mice Epithelial morphology Contain bundles of tonofilaments in the cell cytoplasm and desmosomal regions were prominent at cell boundaries.
HSC2	Oral	TP53 mutation or PIK3CA mutation	Epithelial-like
CAL-27	Tongue	TRET or TP53 mutation	Epithelial, polygonal with highly granular cytoplasm Tumourigenic in nu/nu mice Aneuploid Tumourigenic; solid tumours developed within 6 weeks in nude mice inoculated with 2 × 106 cells subcutaneously Immunocytochemical studies show strong positive staining with antikeratin antibodies
SAS	Tongue	TP53 mutation	Tumourigenic Epithelial-like General cell growth properties
HEp-2	Larynx	Contaminated with HeLa marker chromosome Contains HPV DNA sequence	Epithelial morphology

NR, Not Reported.

**Table 3 biomolecules-12-00290-t003:** Main milk-derived proteins and complexes identified in the included studies.

Protein/Peptide	Structure (Obtained from [19])	Weight (kDa)	Solubility
Lactoferrin	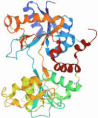	~80 kDa	Soluble
Lactoperoxidase	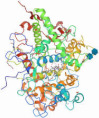	~78 kDa	Soluble
Alpha-lactalbumin	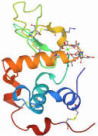	~14 kDa	Soluble
BAMLET/HAMLET	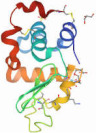	~110 kDa	Soluble

## Data Availability

Data is contained within the article.

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
