# Peer review of "Milk-Derived Proteins and Peptides in Head and Neck Carcinoma Treatment"

_biomolecules, 2022, doi:10.3390/biom12020290_

Round 1
Reviewer 1 Report
This manuscript systematically reviews the current literature on proteins and peptides derived from milk for the treatment of head and neck carcinoma. The report is transparent in how literature was screened, well written and concise. It should be published with minor changes.
In vitro and in vivo should be formatted italic.
The quality of the images (image resolution) should be improved for the final version of the manuscript.
The Table 1 is informative but difficult to read due to the relatively large amount of running text. Please consider reorganization. Please do not use full sentences in a table.
Please add spaces between measurement values and units (whole manuscript).
e.g. “…(tumoral injection of 3-4 doses of 250µg LF for 4 days into C3H/HeJ mice with SCCVII or…” (line 288)
Is the following correct?
“Supplementary Materials: The following are available online at (line 368)
www.mdpi.com/xxx/s1, Figure S1: title, Table S1: title, Video S1: title. (line 369)”
I suggest adding two additional figures:
- Brief visual overview of the substances discussed
- Brief visual overview of the pathways discussed
Author Response
Comment: This manuscript systematically reviews the current literature on proteins and peptides derived from milk for the treatment of head and neck carcinoma. The report is transparent in how literature was screened, well written and concise. It should be published with minor changes.
Reply: Thank you for your comment. The authors are delighted to know that this reviewer appreciated our manuscript.
Comment: In vitro and in vivo should be formatted italic.
Reply: Thank you for your comment. “in vitro” and “in vivo” fonts have been changed throughout the text accordingly.
Comment: The quality of the images (image resolution) should be improved for the final version of the manuscript.
Reply: Thank you for your valuable comment. We improved the quality of Figures 1 and 2 to higher ( at least 600) DPI.
Comment: The Table 1 is informative but difficult to read due to the relatively large amount of running text. Please consider reorganisation. Please do not use full sentences in a table.
Reply: Thank you for your valuable comment. We thoroughly revised and modified Table 1 according to your suggestions. It has been condensed from 4 pages down to 2.5 pages.
Comment: Please add spaces between measurement values and units (whole manuscript). e.g. “…(tumoral injection of 3-4 doses of 250 µg LF for 4 days into C3H/HeJ mice with SCCVII or…” (line 288)
Reply: Thank you for your comment. We added the spacing for units as per your suggestion.
Comment: Is the following correct? “Supplementary Materials: The following are available online at (line 368) www.mdpi.com/xxx/s1, Figure S1: title, Table S1: title, Video S1: title. (line 369)”
Reply: Thank you for your comment. We removed the indicated template sentence as suggested.
Comment: I suggest adding two additional figures:
- Brief visual overview of the substances discussed
- Brief visual overview of the pathways discussed
Reply: Thank you for your comment. While a comprehensive representation of the main pathways was not achievable due to space constraints and lack of consistent reporting across all studies, these authors feel that a concise visual overview of the main proteins and protein complexes would add value to the manuscript. We added the latter per your request and it currently reads Table 1.
Reviewer 2 Report
The manuscript of Liu X et al. reviews the anti-cancer effects of proteins and peptides derived from milk observed in vitro on cell lines derived from head and neck squamous cell carcinoma (HNSCC) and more rarely on murine models (one study of 8 reviewed). This research topic presents a real interest for biomedical field and the research questions are well addressed by the authors of this manuscript. Unfortunately, the authors fail to respond properly to these questions, which renders the manuscript less interesting and scientifically sound. Several suggestions are proposed below to the authors in order to improve their manuscript.
Major corrections
Title of the manuscript:
- The title suggests that milk derivatives may treat HNSCC, but no clinical studies are presented in this manuscript. The reviewed publications present studies performed preclinically exclusively, so the title should be adapted.
Materials and Methods:
- The inclusion criteria (chapter 2.1.1) used by the authors to select the reviewed publications were based on PICOS tool questions, which are appropriate for clinical studies, but not for preclinical and especially in vitro
- Regarding the exclusion criteria (chapter 2.1.2), the 3rd and the 4th ones would be interesting to be included; they may help to answer the research questions.
- In chapter 2.1.3, the authors refer to the Supplementary Table 1, where the syntaxes employed for literature search are listed. This reviewer could not find this table. The authors may refer to Appendix A. The Supplementary Table 2 mentioned in chapter 2.3 could not be found either. The authors may refer to Appendix B.
- Among the 658 screened papers, the authors retained only 8 of them, which is not enough to formulate confident conclusions for a review manuscript.
- It would be useful to present a list of cell lines used as models of so many heterogenous and different types of HNSCC. If presented within the manuscript, their tissue of origin, genetic characteristics and the specific mutations would be very informative and may help the authors to draw conclusions regarding the sensitivity/resistance to this kind of treatment of both cancer and healthy cells.
- The QUIPS (quality in prognosis studies) tool employed in this manuscript to evaluate the risk of bias is appropriate for clinical studies, but not for this kind of review research done on preclinical studies.
Results:
- The results of this review research are presented in a simplified manner. The main ideas are resumed in Table 1, which is quite difficult to follow up because of the condensed information. It would be more appropriate to present them clearly, completely and critically within the main text of the manuscript.
- As explained above, the inclusion and exclusion criteria do not seem appropriate for the present study, so the study selection (chapter 3.1) should be revised.
- The countries where these studies were performed are not definitory for the study characteristics (chapter 3.1.1).
- Again, the QUIPS tool is not suitable for this study, as explained above.
- The complete name and function of various genes and the expressed proteins should be explained, as well as the signaling pathways that may be modulated and/or affected by milk derivatives. This will allow to conceive theories regarding their mechanisms of action.
- The listing of specific mutations in different HNSCC cell lines could help explaining the reason of their different reactivity to milk derivatives as compared to healthy cells.
Discussion:
- What signaling molecules involved in autophagy, cell cycle arrest and apoptosis were modulated by milk derivatives?
- What kind of morphological changes were observed under the effect of milk derivatives?
- What scientific evidences sustained the immunomodulation by milk derivatives?
- What scientific evidences demonstrated that milk proteins reduce the levels of transcription factors? What transcription factors were affected?
- The authors could conclude and propose mechanisms of action for milk proteins, but also paths for future investigations.
Minor corrections
- All abbreviations should be explained when first introduced in the text.
- The authors should double check the spelling and grammar.
Author Response
Comment: The manuscript of Liu X et al. reviews the anti-cancer effects of proteins and peptides derived from milk observed in vitro on cell lines derived from head and neck squamous cell carcinoma (HNSCC) and more rarely on murine models (one study of 8 reviewed). This research topic presents a real interest for the biomedical field and the research questions are well addressed by the authors of this manuscript. Unfortunately, the authors fail to respond properly to these questions, which renders the manuscript less interesting and scientifically sound. Several suggestions are proposed below to the authors in order to improve their manuscript.
Reply: Thank you for this comment, the authors are delighted to hear that the research questions were well addressed by these authors.
Major corrections
Title of the manuscript:
Comment: The title suggests that milk derivatives may treat HNSCC, but no clinical studies are presented in this manuscript. The reviewed publications present studies performed preclinically exclusively, so the title should be adapted.
Reply: Thank you for your comment. These authors acknowledge that the systematic review process did culminate in the inclusion of 8 preclinical studies. However, since the inception of the design, the authors aimed to include all types of original studies in the review as stated in the inclusion criteria. This is sustantiated by our inclusion criteria listed on page 3, line 78. However, due to the nature of the published literature available to date, we were unable to find any clinical studies. As such, the authors feel that this title most accurately represents the intention of the study from the outset.
Materials and Methods
Comment: The inclusion criteria (chapter 2.1.1) used by the authors to select the reviewed publications were based on PICOS tool questions, which are appropriate for clinical studies, but not for preclinical and especially in vitro.
Reply: Thank you for your comment. As mentioned above our aim was not to exclude clinical studies, but rather to conduct a comprehensive search of all available literature. This intent was clearly stated on page 3, line 78, by our inclusion and exclusion criteria. However, due to the nature of the published literature available, no clinical studies that fit our criteria were identified. As such, the use of the PICOS tool was appropriate in this case as the authors aimed to include all types of studies in the review from the outset, utilising the “P” of PICOS to refer to patient/population/case in order to ensure the tool was also applicable to preclinical studies.
As stated by the Cochrane institute, “The criteria for considering types of people included in studies in a review should be sufficiently broad to encompass the likely diversity of studies and the likely scenarios in which the interventions will be used, but sufficiently narrow to ensure that a meaningful answer can be obtained when studies are considered together they should be specified in advance….As discussed in Chapter 2, Section 2.3.1, the degree of breadth will vary, depending on the question being asked and the analytical approach to be employed. A range of evidence may inform the choice of population characteristics to examine, including theoretical considerations, evidence from other interventions that have a similar mechanism of action, and in vitro or animal studies…”
Please feel free to refer to the following link: https://training.cochrane.org/handbook/current/chapter-03
Comment: Regarding the exclusion criteria (chapter 2.1.2), the 3rd and the 4th ones would be interesting to be included; they may help to answer the research questions.
Reply: Thank you for your comment.
In regard to the exclusion criteria specified in the 3rd point, historically these cancers are not considered head and neck cancer and are instead classified as separate anatomical regions.
Please feel free to refer to: https://www.cancer.gov/types/head-and-neck/head-neck-fact-sheet.
Inclusion of these additional anatomical areas would require reformulation of our study design and is therefore be considered infeasible and outside the scope of interest for this systematic review. Further research in these different anatomical areas may be addressed by other researchers in the near future.
Comment: In chapter 2.1.3, the authors refer to the Supplementary Table 1, where the syntaxes employed for literature search are listed. This reviewer could not find this table. The authors may refer to Appendix A. The Supplementary Table 2 mentioned in chapter 2.3 could not be found either. The authors may refer to Appendix B.
Reply: Thank you for this valuable comment. We identified the discrepancy and addressed this point accordingly.
Comment: Among the 658 screened papers, the authors retained only 8 of them, which is not enough to formulate confident conclusions for a review manuscript.
Reply: Thank you for your valuable comment. The authors strongly agree with this reviewer that the inclusion of only 8 papers provides limited evidence from which conclusions were able to be drawn. Furthermore, the limited number of studies that fulfilled our inclusion criteria precluded quantitative synthesis of the findings, such as meta-analysis. This is a key limitation of our review and as such was outlined on line 431. It is important to note that routine systematic review protocols were followed in the selection of studies for inclusion in this review. As such, the small number of studies that would ultimately be included in the review could not have been known a priori.
According to your valuable comment, the authors edited this section of the discussion, further emphasising these limitations.
Comment: It would be useful to present a list of cell lines used as models of so many heterogenous and different types of HNSCC. If presented within the manuscript, their tissue of origin, genetic characteristics and the specific mutations would be very informative and may help the authors to draw conclusions regarding the sensitivity/resistance to this kind of treatment of both cancer and healthy cells.
Reply: Thank you for this valuable comment. Following your valuable suggestion, we added a table summarising the cell lines identified. This table now reads table number 1.
Comment: The QUIPS (quality in prognosis studies) tool employed in this manuscript to evaluate the risk of bias is appropriate for clinical studies, but not for this kind of review research done on preclinical studies.
Reply: Thank you for your comment. We agree that the QUIPS tool was not the most appropriate tool, despite this has been widely used in the literature for in vitro studies. As such the authors decided to perform a completely new analysis of the risk of bias using the most appropriate tool, OHAT, which perfectly aligns with the nature of our study. Please feel free to refer to table 1 at https://www.nhmrc.gov.au/guidelinesforguidelines/develop/assessing-risk-bias.
Our new risk of bias assessment is located on page 4, line 127 within our manuscript draft.
Results
Comment: The results of this review research are presented in a simplified manner. The main ideas are resumed in Table 1, which is quite difficult to follow up because of the condensed information. It would be more appropriate to present them clearly, completely and critically within the main text of the manuscript.
Reply: Thank you for your comment. The amount of data included in the 8 studies was large, therefore the table must be condensed for space constraints (as pointed out by reviewer 1). However, we also agree with this reviewer that the results must be critically appraised in our review and, as such, used the discussion section for this purpose, presenting the literature in a more complete and critical manner. All the corrections for the discussion section are visible using track-changes.
Comment: As explained above, the inclusion and exclusion criteria do not seem appropriate for the present study, so the study selection (chapter 3.1) should be revised.
Reply: Thank you for your comments. As discussed in relation to your previous comments, the PICOS design is robustly applicable to our study design. These authors feel that the exclusion and inclusion criteria are well designed for the purposes of this review and are tailored to our specific research questions.
Comment: The countries where these studies were performed are not definitory for the study characteristics (chapter 3.1.1).
Reply: Thank you for these valuable comments. We strongly agree with you that the countries are not definitory for the study characteristics of this review, and as such the countries have been promptly removed.
Comment: Again, the QUIPS tool is not suitable for this study, as explained above.
Reply: Thank you for this comment. Following this reviewer’s previous concern regarding the use of the QUIPS tool within the manuscript, this has already been removed.
Comment: The complete name and function of various genes and the expressed proteins should be explained, as well as the signalling pathways that may be modulated and/or affected by milk derivatives. This will allow us to conceive theories regarding their mechanisms of action.
Reply: Thank you for this valuable comment. We added to the discussion a paragraph that shed further light on the pathways involved. The discussion has been modified accordingly to address this request.
Comment: The listing of specific mutations in different HNSCC cell lines could help explain the reason for their different reactivity to milk derivatives as compared to healthy cells.
Reply: Thank you for this valuable comment. We completely agree with this reviewer and the specific mutations for each cell line have been described in table 1. We further analysed these links throughout the discussion section.
Discussion:
Comment: What signalling molecules involved in autophagy, cell cycle arrest and apoptosis were modulated by milk derivatives?
Reply: Thank you for your comment. As per your request, we have added information regarding possible signalling molecules involved to the discussion.
Comment: What kind of morphological changes were observed under the effect of milk derivatives?
Reply: Thank you for your comment. The most common morphological changes identified were membrane blebbing, cellular shrinkage, apoptotic bodies, nuclear condensation and fragmentation of the DNA, identified by Mohan et al. (2007) and Sakai et al. (2005). The discussion has been modified accordingly to address this request.
Comment: What scientific evidence sustained the immunomodulation by milk derivatives?
Reply: Thank you for your comment. Evidence of immunomodulation was identified by Wolf et al. (2003), demonstrating the ability of lactoferrin to upregulate the host cellular immune system. The discussion has been modified accordingly to address this request, including further discussion of immunomodulation.
Comment: What scientific evidence demonstrated that milk proteins reduce the levels of transcription factors? What transcription factors were affected?
Reply: Thank you for your comment. From the 8 papers included in this review, there was limited evidence regarding the specific role of transcription factors. For example, Panahipour et al. (2019) were unable to definitively determine whether the change in transcript factor would translate to a cellular response. This was identified as a limitation which the authors addressed as a potential avenue for future research.
Comment: The authors could conclude and propose mechanisms of action for milk proteins, but also paths for future investigations.
Reply: Thank you for your comment. As per your suggestion, the authors broadened the section of the discussion specifically dedicated to paths for future investigation, emphasising several possible avenues including the need for future research that utilises scientifically sound protocols as well as investigating the effect of milk-derived proteins in combination with established chemotherapeutic agents.
Minor corrections
Comment: All abbreviations should be explained when first introduced in the text.
Reply: Thank you for this valuable comment. We critically revised the use of abbreviations throughout the entire manuscript, ensuring each one was explained when first introduced in the text.
Comment: The authors should double-check the spelling and grammar.
Reply: Thank you for your comment. The authors thoroughly revised all the spelling and grammar errors within the article.
Round 2
Reviewer 2 Report
The manuscript of Liu X et al. has significantly been improved based on the reviewers’ suggestions. Several revisions are furthermore required before accepting the manuscript for publication in Biomolecules journal.
Major corrections
Results:
- Chapter 3.3, L203: Murine is a family of rodents that includes mice and rats. Therefore, the sentence “2 murine mice models and 1 mice model” is redundant and not scientifically sound. Therefore, these animal models were either 3 mouse models or 3 murine models or 2 rat and 1 mouse model. Please correct accordingly.
- Table 1: The title requires correction to reflect its content, eventually as follows: “Properties of HNSCC cell lines: tissue of origin, genetic characteristics and mutations, and tumorigenic potential”. The head of the last column could be titled “Tumorigenic properties”. However, its content is not very clear. For instance, one may understand that O22 (JHU-O22) cell line is not tumorigenic or that this property is not reported. However, this cell line produces laryngeal squamous cell carcinoma. The O12 (JHU-O12) cell line was shown to be a derivative of JHU-O22; it also produces laryngeal squamous cell carcinoma. Ca9-22 cell line produces gingival squamous cell carcinoma. SAS cell line produces tongue squamous cell carcinoma. The cell lines listed in this table are not all of them metastatic? Some of them, but not all, are producing cancer in mice? The morphological aspects are described for CAL-27 cell line, but not for the others. They are not available?
- The title of Table 2 should also be corrected to reflect the content. Is this table showing protein complexes, or it comprises milk-derived proteins and peptides?
- The title of Table 3 needs also to be corrected. The authors may mean “… proteins and peptides on head and neck cancer cells”.
Discussion:
- L331-340: The studies described in this paragraph do not correspond to the content of reference 21. This reviewer could not verify their validity. Also, endoplasmic reticulum and proteasomes (L340) are two different cellular entities, unless the author refer to some specific proteasomes associated to endoplasmic reticulum (which ones?).
- L341-357: This paragraph does not correspond either with reference 22. There seems to be a general mismatch of references. The authors are invited to check the scientific content and references within the entire manuscript. Moreover, ERK1/2 phosphorylation should not induce apoptosis. The authors are invited to double check this mechanistic information.
- L382: The results of Panahipour (ref. 21 instead of 23) suggest (instead of demonstrate) the reduced expression of transcription factors.
- L412-414: The sentence is at the limit of comprehension and needs to be rewritten.
- L420: What is the meaning of “titration of calcium-free-hLA-free by OA”?
- L423: What is the meaning of “a more transparent manufacturing process”?
- L434: The authors may probably mean that cytotoxic effects occur on cancer cells (i.e., not on healthy ones).
- L442: Why the synthesis of milk-derived proteins would be complex?
Minor corrections
- L199-200: Cell line O12 and O22 instead of o12 and o22
- L312-314: The adverb “however” appears two times in the same phrase. Please consider phrase reformulation.
- L313, L316 and L319 (and elsewhere): in vitro and in vivo should be italic
- L314: IC50 should have “50” in subscript (IC50)
- L317: “it significantly”
- L318: “mg/ml of lactoferrin”
- L345: “treatment with bovine”
- L352: “cotreatment with P-E and LF”
- L360: “intra-tumoral injection of 250 µg”
- L364: “by up-regulation”
- L368: “proteins used, a correlation”
- L375: angiogenesis, while”
- L378: “known to bind”
- L384: “proteins” instead of “protein”
- L422: What is the meaning of CMC?
- L425: “review, several limitations”
- L428: “studies, the potential”
- L430: “treatment with milk-derived”
- L432: “truly are selectively” (please delete “at”)
Author Response
The manuscript of Liu X et al. has significantly been improved based on the reviewers’ suggestions. Several revisions are furthermore required before accepting the manuscript for publication in Biomolecules journal.
MAJOR CORRECTIONS
Results
Comment: Chapter 3.3, L203: Murine is a family of rodents that includes mice and rats. Therefore, the sentence “2 murine mice models and 1 mice model” is redundant and not scientifically sound. Therefore, these animal models were either 3 mouse models or 3 murine models or 2 rat and 1 mouse model. Please correct accordingly.
Reply: Thank you for your valuable comment, we have corrected the text accordingly.
Comment: Table 1: The title requires correction to reflect its content, eventually as follows: “Properties of HNSCC cell lines: tissue of origin, genetic characteristics and mutations, and tumorigenic potential”. The head of the last column could be titled “Tumorigenic properties”. However, its content is not very clear. For instance, one may understand that O22 (JHU-O22) cell line is not tumorigenic or that this property is not reported. However, this cell line produces laryngeal squamous cell carcinoma. The O12 (JHU-O12) cell line was shown to be a derivative of JHU-O22; it also produces laryngeal squamous cell carcinoma. Ca9-22 cell line produces gingival squamous cell carcinoma. SAS cell line produces tongue squamous cell carcinoma. The cell lines listed in this table are not all of them metastatic? Some of them, but not all, are producing cancer in mice? The morphological aspects are described for CAL-27 cell line, but not for the others. They are not available?
Reply: Thank you for the comments and examples. We have made changes to address all these concerns, such that the title reflects the content better.
Comment: The title of Table 2 should also be corrected to reflect the content. Is this table showing protein complexes, or it comprises milk-derived proteins and peptides?
Reply: Thank you for your comment. We have changed the title to “Main milk derived proteins and complexes identified in the included studies” as there are no peptides in the table.
Comment: The title of Table 3 needs also to be corrected. The authors may mean “… proteins and peptides on head and neck cancer cells”.
Reply: Thank you for your comment. The title has been changed to “… proteins and peptides on head and neck cancer cells”.
Discussion
Comment: L331-340: The studies described in this paragraph do not correspond to the content of reference 21. This reviewer could not verify their validity. Also, endoplasmic reticulum and proteasomes (L340) are two different cellular entities, unless the author refer to some specific proteasomes associated to endoplasmic reticulum (which ones?).
Reply: Thank you for your comment, we changed the references to reflect the information more accurately.
Comment: L341-357: This paragraph does not correspond either with reference 22. There seems to be a general mismatch of references. The authors are invited to check the scientific content and references within the entire manuscript. Moreover, ERK1/2 phosphorylation should not induce apoptosis. The authors are invited to double check this mechanistic information.
Reply: Thank you for pointing out the reference mismatch. We have now fixed it and all the content should match to their corresponding references. We do not mention that ERK1/2 phosphorylation induces apoptosis but instead plays a “play a key role in cell growth, survival, differentiation, and apoptosis. We changed the wording to make the following sentence clearer.
Comment: L382: The results of Panahipour (ref. 21 instead of 23) suggest (instead of demonstrate) the reduced expression of transcription factors.
Reply: Thank you for your comment. We have made amendments accordingly.
Comment: ML412-414: The sentence is at the limit of comprehension and needs to be rewritten.
Reply: Thank you for your comment. We have rewritten the whole sentence in question to be more comprehensive.
Comment: What is the meaning of “titration of calcium-free-hLA-free by OA”?
Reply: Thank you for your comment. We have added more information about the titration in the paper.
Comment: L423: What is the meaning of “a more transparent manufacturing process”?
Reply: Thank you for your comment. This phrase is used by the original authors to describe a preparation process that is clearer that allows for modification within the steps. We have deleted ‘transparent’ to avoid confusion
Comment: L434: The authors may probably mean that cytotoxic effects occur on cancer cells (i.e., not on healthy ones).
Reply: Thank you for this comment. We corrected this misleading sentence using “normal” where appropriate.
Comment: L442: Why the synthesis of milk-derived proteins would be complex
Reply: Thank you for this comment. This phrase was referring to protein-complexes as opposed to the complexity of synthesis. We have edited this sentence to reflect this.
MINOR CORRECTIONS
- L199-200: Cell line O12 and O22 instead of o12 and o22
- L312-314: The adverb “however” appears two times in the same phrase. Please consider phrase reformulation.
- L313, L316 and L319 (and elsewhere): in vitro and in vivo should be italic
- L314: IC50 should have “50” in subscript (IC50)
- L317: “it significantly”
- L318: “mg/ml of lactoferrin”
- L345: “treatment with bovine”
- L352: “cotreatment with P-E and LF”
- L360: “intra-tumoral injection of 250 µg”
- L364: “by up-regulation”
- L368: “proteins used, a correlation”
- L375: angiogenesis, while”
- L378: “known to bind”
- L384: “proteins” instead of “protein”
- L422: What is the meaning of CMC?
- L425: “review, several limitations”
- L428: “studies, the potential”
- L430: “treatment with milk-derived”
- L432: “truly are selectively” (please delete “at”)
Reply: Thank you for the comments, all the above changes have been made. According to this reviewer requests.